# Four Faces of Cell-Surface HLA Class-I: Their Antigenic and Immunogenic Divergence Generating Novel Targets for Vaccines

**DOI:** 10.3390/vaccines10020339

**Published:** 2022-02-21

**Authors:** Mepur H. Ravindranath, Narendranath M. Ravindranath, Senthamil R. Selvan, Edward J. Filippone, Carly J. Amato-Menker, Fatiha El Hilali

**Affiliations:** 1Department of Hematology and Oncology, Children’s Hospital, Los Angeles, CA 90027, USA; 2Emeritus Research Scientist at Terasaki Foundation Laboratory, Santa Monica, CA 90064, USA; 3Norris Dental Science Center, Herman Ostrow School of Dentistry, University of Southern California, Los Angeles, CA 90089, USA; nravindr@usc.edu; 4Tetracore Inc., Rockville, MD 20850, USA; senthamils59@gmail.com; 5Division of Nephrology, Department of Medicine, Sidney Kimmel Medical College, Thomas Jefferson University, Philadelphia, PA 19145, USA; kidneys@comcast.net; 6Department of Microbiology, Immunology and Cell Biology, School of Medicine, West Virginia University, Morgantown, WV 26506, USA; cjc0082@mix.wvu.edu; 7The Faculty of Medicine and Pharmacy of Laayoune, Ibn Zohr University, Agadir 70000, Morocco; f.elhilali@ulz.ac.ma

**Keywords:** HLA, heavy chain, α1 and α2 helices, β2microglobulin, monomeric, homodimers, heterodimers, monoclonal antibodies, Face 1, Face 2, Face 3, Face 4

## Abstract

Leukocyte cell-surface HLA-I molecules, involved in antigen presentation of peptides to CD8+ T-cells, consist of a heavy chain (HC) non-covalently linked to β2-microglobulin (β2m) (Face-1). The HC amino acid composition varies across all six isoforms of HLA-I, while that of β2m remains the same. Each HLA-allele differs in one or more amino acid sequences on the HC α1 and α2 helices, while several sequences among the three helices are conserved. HCs without β2m (Face-2) are also observed on human cells activated by malignancy, viral transformation, and cytokine or chemokine-mediated inflammation. In the absence of β2m, the monomeric Face-2 exposes immunogenic cryptic sequences on these cells as confirmed by HLA-I monoclonal antibodies (LA45, L31, TFL-006, and TFL-007). Furthermore, such exposure enables dimerization between two Face-2 molecules by SH-linkage, salt linkage, H-bonding, and van der Waal forces. In HLA-B27, the linkage between two heavy chains with cysteines at position of 67 of the amino acid residues was documented. Similarly, several alleles of HLA-A, B, C, E, F and G express cysteine at 67, 101, and 164, and additionally, HLA-G expresses cysteine at position 42. Thus, the monomeric HC (Face-2) can dimerize with another HC of its own allele, as homodimers (Face-3), or with a different HC-allele, as heterodimers (Face-4). The presence of Face-4 is well documented in HLA-F. The post-translational HLA-variants devoid of β2m may expose several cryptic linear and non-linear conformationally altered sequences to generate novel epitopes. The objective of this review, while unequivocally confirming the post-translational variants of HLA-I, is to highlight the scientific and clinical importance of the four faces of HLA and to prompt further research to elucidate their functions and their interaction with non-HLA molecules during inflammation, infection, malignancy and transplantation. Indeed, these HLA faces may constitute novel targets for passive and active specific immunotherapy and vaccines.

## 1. Introduction

### 1.1. Preamble

A protein can be a monomeric single polypeptide chain or two or more such chains (e.g., dimeric or trimeric). A monomeric protein may have secondary structures (α-helix and β-pleated sheets) (Figure 1) and attain a tertiary structure. However, when proteins become dimers or trimers they attain a quaternary structure.

The amino acid sequences or epitopes exposed on a single polypeptide heavy chain may become cryptic when the protein attains a quaternary structure. Consequently, the antigenicity and immunogenicity of the same protein may differ between the single heavy chain and the same protein having a quaternary structure. If the protein is a glycosylated polypeptide, the exposure of the sugar residues on the protein may differ between the single heavy chain version and its di- or trimeric quaternary version. Protein glycosylation occurs in the endoplasmic reticulum (ER) and Golgi apparatus [1]. In multiple enzymatic steps, a complex oligosaccharide (i.e., glycan) is synthesized in the ER, transferred to a specific receptor sequence on its target protein, and subsequently cropped and remodeled in the ER and Golgi apparatus. 

### 1.2. Polymorphism of Human Leukocyte Antigens (HLA)

The genes encoding heavy chains (HC) of HLA class-I and HLA class-II are located on the short arm of human chromosome 6. HLA-I genes are located near the telomere, whereas HLA-II genes are located near the centromere. The six isoforms of HLA-I are positioned in the following sequence: Centromere_HLA-II_HLA-B--HLA-C--HLA-E--HLA-A--HLA-G—HLA-F-Telomere

Glycosylated HLA molecules predominantly occur on the cell surface as quaternary structures with a propensity to bind with peptides. HLA-I molecules that are expressed on the cell surface occur as trimers consisting of a HC, non-covalently linked β2-microglobulin (β2m), and a peptide, usually 8–10 amino acids in length, embedded in the polymorphic binding groove of the HC (Figure 2). The β2m is the β chain of HLA-I and the β2m gene is located on chromosome 15. The β2m is monomorphic in contrast to the polymorphic HLA-HCs. These heterodimers of HLA-I, also known as closed conformers [2], are designated as Face-1. The antigenic polymorphism of HLA class-I isoforms is primarily due to the inter- and intra-isoform differences in the amino acid sequences of the HC. Phylogenetically, HLA-HC alone was reported in vertebrates including *Ambystoma*, fishes, clawed toad, turtle, snake, and caiman, and HC with β2-m in clawed toad, chicken and other mammals [3,4].

The primary role of both HLA-I and HLA-II is to present peptides to CD8+ and CD4+ T lymphocytes, respectively [2,5]. Accordingly, the T-cell receptors, CD8 and CD4, interact with HLA-I and HLA-II molecules. The site of interaction of CD8 with class-I molecules is the α3 domain, and that of CD4 with class-II is the β2 domain. Presentation of peptides by HLA-I to CD8+ T-cells results in the stimulation of the CD8+ cytotoxic T cells (CTLs) to kill cells infected with viruses and intracellular pathogens. The bound peptides are derived from intracellular proteins (endogenous) or viral polypeptides produced upon viral infection that have been cleaved by the proteasome and other cytoplasmic peptidases. These peptides are transported into the ER, usually by the transporters associated with antigen processing (TAPs). In the ER, the peptides are loaded onto the groove of the HLA-I heavy chain. The peptide carried by Face-1 is thus expressed on the cell surface. After presenting the peptide or when peptide is shed, the β2m dissociates itself from the HC and is released into the microenvironment of the cell and then into the circulation. The shedding is facilitated by calcium dependent and zinc-containing matrix metalloproteinases (MMPs), such as MMP-2, MMP-3, MMP-8 and MMP-9 [6,7,8]. The HLA heavy chains and β2m are degraded soon after release into the circulation.

HLA-II presents peptides from exogenous proteins to CD4+ T helper cells, which deliver cytokines to immune cells resulting in their activation. Both CD4+ and CD8+ T cells contribute to adaptive immunity, which can be enhanced by vaccination. The polymorphic differences in the α1 and α2-helical structures among the HLA molecules impact both the nature of the peptide antigens presented as well as the nature of the responding T cells [9]. Amino acid sequencing of thousands of HLA-I and -II molecules revealed that the sequence diversity is concentrated at specific regions of the HLA molecule. A T-cell receptor (TcR) interacts with the three-dimensional structure of a peptide carried on an HLA class-I or -II molecule.

The specificity of TcR restriction is a consequence of the sequence diversity of the HLA molecule. Such sequence diversity has generated wide varieties of HLA-I and -II molecules. The number of named alleles for each HLA-I and HLA-II gene and their respective proteins were reported recently (7 July 2021) in the IPD-IMGT/HLA DATABASE (Table 1).

### 1.3. Natural Cell-Surface HLA Molecules Are Glycosylated 

In normal cells, HLA-I have fucosylated biantennary structures, while HLA-DR (HLA-II) have bi-, tri- and tetra-antennary and high-mannose structures. On HLA I, the position *N*86 (Asparagine on 86th amino acid) is glycosylated. The *N*86 is located close to the peptide-binding groove [10]. It is highly conserved and present on practically all HLA-I allotypes [11,12,13]. The HLA-II α chain of Face-I contains both an N- linked high-mannose carbohydrate chain and a complex-type carbohydrate chain. The HLA-II β chain of Face-1 contains only an N-linked, complex-type carbohydrates chain. Several viruses, such as influenza, express neuraminidase, which may remove the terminal sialic acids (NeuAc, NeuGc, 9-O-NeuAc, 9-O-NeuGc) of these N-linked glycans and change their net negative charge caused by such sialic acids. 

The glycosylation residues may differ by cell types within a person [14,15]. For example, the number of HLA-II is higher in B cells than in monocytes. The pattern of glycosylation in B cells upon infection with Epstein-Barr virus (EBV) differs from native B cells. HLA-I molecules in the EBV- transformed B-cell line are under-sialylated, whereas HLA-II molecules are over-sialylated. Both tri-antennary and high-mannose structures are increased in HLA-I molecules. Cytomegalovirus infection can also induce changes in HLA glycosylation [16]. The pattern of glycosylation of Face-1 on the surface of malignant human cancer cells differs markedly from that of their non-malignant counterparts. Yoneyama et al. [17] report a novel mechanism governing HLA class-I (HLA-A24) cell-surface expression in bladder cancer. It involves specifically altered post-translational modification by O-glycosylation of HLA-I. The altered glycosylation of HLA-I enables the bladder cancer cells to metastasize to lymph nodes and to evade cytotoxic T lymphocyte (CTL) immunity. It appears that the glycosyl residues on glycoproteins serve as a checkpoint for proper protein folding and trafficking of Face-1 to the cell surface [18,19,20]. Glycosylation of Face-1 protects it from proteolysis once it is on the cell surface [21] and allows for spacing of receptors and ligands on the cell surface for optimized cell-cell attachment [22]. Similarly, Ryan et al. [23] have demonstrated that the presence of the carbohydrate moiety on HLA-II is essential for presentation of a certain class of peptides. 

In contrast to native glycosylated HLA-DR2, recombinant (bacterially expressed) DR2 failed to associate with peptides despite being properly folded and fully capable of such binding. Perhaps even more significant, alterations in cellular protein glycosylation of both HLA-I and HLA-II have been reported in inflammatory bowel disease and patients with colon cancer [24,25,26]. Thus, it is evident that a single HLA allele is also diversified due to the differences in glycosylation between cell types, before and after infection, inflammation and malignant diseases. These diversifications may impact both the antigenicity and the immunogenicity of HLA.

Monitoring anti-HLA-I antibodies post-transplantation is an essential step in clinical evaluation. HLA-I and HLA-II reactivities of IgG are analyzed by Single Antigen Bead (SAB) assays using dual-laser flow cytometry with Luminex xMAP^®^ multiplex technology (LABScan™ 100). The SABs used to monitor antibodies (e.g., LABScreen, One Lambda Inc., Canoga Park, CA, USA) are coated with recombinant HLA and are not glycosylated. Clinicians deduce the existence of donor specific antibodies (DSA) based on the reactivity of IgG antibodies to the “non-glycosylated version” of HLA-I coated on such beads. Evidently, antibody recognition of the epitopes (at the heart of which are eplets near the binding groove) may differ depending on the presence or absence of glycosylation residues (in vivo). Antibodies recognizing non-glycosylated HLA molecules coated on solid matrices, may not be accessible to the same sites in the presence of glycosylation residues. This could possibly be one of the reasons why DSAs identified by SAB assays may fail to recognize the natural HLA-I and HLA-II molecules on the allograft and, therefore not be detrimental [26,27,28,29,30,31,32,33]. Probably for the same reason, the monoclonal antibodies developed by immunizing deglycosylated recombinant protein molecules (e.g., IL-2), do not target the specific antigen in passive immunotherapy. Interestingly the number of glycosylation sites in HLA-I of normal cells is singular, whereas it is several in HLA-II. Wilson et al. [34], employing a diagnostic mAb (Q6/64), demonstrated that the glycosylation of the HLA-I HC can hinder antibody recognition either directly or by altering the conformation of the α1 helix. There is a need to identify and characterize the mAbs that can distinguish glycosylated from non-glycosylated HCs of HLA-I and -II molecules. Until that is done, the HLA antibodies claimed to be “HLA-allele-specific” or “donor-HLA-allele-specific” will remain confounded.

## 2. The Four Faces of the HLA Class-I Molecules

The HLA-I heterodimers involved in peptide presentation (closed conformers), described above constitute the Face-1 of HLA-I molecules. This structure changes post-translationally on the surface of cells activated by different stimuli. In activated lymphocytes, virally infected cells, and malignant cells, β2-m-free HCs are expressed naturally on the cell surface as monomers, and they are referred to as Face-2. The monomer without β2m may not have a stable grove to present peptides. The monomers may expose cysteines at differ positions of α1, α2 and α3 helices and may get linked with cysteines of other Face-2 molecules on the cell surface by forming S-S bonds depending on the conformation of the heavy chains. Therefore, these monomers may interact with other monomers of the same HLA allele to become homodimers (Face-3) or with monomers of other HLA-I alleles to generate heterodimers (Face-4). Figure 3 diagrammatically illustrates the naturally occurring four faces of HLA-I molecules.

### 2.1. Face-2: The Open Conformers

Early studies [34,35,36,37] documented that the Daudi Burkitt lymphoma cell line expressed neither HLA-I antigens nor β2m on the cell surface. The primary defect in this cell line is the inability to synthesize β2m, as shown by biochemical studies and by somatic cell hybridization experiments. However, the expression of Daudi HLA-I antigens could be rescued either by mouse or human β2m, provided by the normal counterpart of Daudi. Strikingly similar data were obtained using a somatic cell variant of the C3H (H-2k) lymphoma. Based on these findings, it was concluded that the heavy chains of class-I molecules cannot be expressed in the absence of β2m.

As early as 1979, the Strominger team [38,39] used mouse anti-HLA mAb W6/32 to identify intact HLA-I (Face-1). The mAb recognized an epitope on the β2m residues 3 and 89, and on the HC residue 121 of all isoforms of intact HLA (Face-1) but not β2m-free HC (Face-2) [40,41]. They studied the biosynthesis of HLA-I in the human B lymphoblastoid cell line T5-1, which is positive for HLA-Al, -A2, -B8, and -B27, after immunoprecipitation of the detergent lysates of ^35^S-methionine-labeled cells and testing with W6/32. Indeed, the mAb recognized specifically the Face-1. They also immunoprecipitated the lysate with an anti-H polyclonal serum, which does not bind to Face-1, but only the β2m-free HLA (Face-2). The results confirmed the presence of β2m-free HLA (Face-2). Based on these findings, they reported that “*W6/32 reactive material was detected on the cell surface, whereas anti-H-reactive material … could only sometimes be detected in small quantities on overexposed fluorographs. Thus, at most only 1 or 2% of the heavy chains (Face-2) present on the surface of T5-1 were precipitable by anti-H*” ([38], p. 984).

However, Nossner and Parham [42], emphasized the need for conclusive evidence to support the expression of Face-2 of HLA on the cell surface of human cells. They have summarized previous reports [43,44,45,46,47] that provided circumstantial evidence for differences in the biogenesis of HLA class-I molecules in human and mouse cells. In mouse cell lines that lack expression of β2m, some fraction of Face-2 reaches the cell surface which is also true for mice with genetically knocked out β2m genes. But in the Daudi human cell line, which is defective in β2m, HLA-I molecules are trapped in the ER and appear to be actively prevented from reaching the cell surface [48], as discussed above. Since this investigation, cell surface expression of Face-2 received focused attention, leading some investigators to speculate that Face-2 of HLA could be immunologically inert [49], which was shown to be incorrect [50,51,52,53,54,55,56]. In any case, expression of Face-2 is observed across lymphocyte subpopulations (CD3+ T cells, CD19+ B cells, CD56+ NK cells and CD14+ monocytes), and importantly these HCs are overexpressed particularly on activated cells, as well as on extravillous trophoblast cells and monocytes [57]. The origin and functions of Face-2 in activated cells have been speculated on in these investigations.

### 2.2. The mAb LA45 Recognizes Only Face-2 but Not Face-1

Five years after the Strominger group reports [38,39], Allen et al. [44] demonstrated that the transfected murine class-I molecule H-2D^b^ can be expressed on the surface in the absence of β2m (Face-2). These free HCs (Face-2s) display a significantly altered conformation unrecognized by allospecific and Db-restricted CTLs or by most alloantibodies. Schnabl et al. [58] observed β2m-free HLA-I (Face-2) on the cell surface of both in vitro and in vivo activated human T lymphocytes, but not on the resting ones. In naïve T-cells, the β2m in HLA-I is noncovalently associated with some amino acids in the α3-subunit of the heavy chain [59]. The conformational properties of Face-2 of HLA-I are found to be very different from peptide binding Face-1. They performed sequential immunoprecipitation with mAb W6/32 and LA45 (raised against HTLV-1 transformed human T cell line HUT102) to study the Face-2. W6/32 precipitated Face-1, whereas LA45 precipitated only Face-2 and did not co-precipitate β2m. Western blot analysis of the cell lysates further confirmed the results.

Further investigation with mAb LA45 on freshly isolated mononucleated immune cells of 18 volunteers confirmed the presence of weak cell surface expression of the LA45 positive antigen. After phytohaemagglutinin (PHA) activation of the T-lymphocytes from 12 healthy individuals, the cell surface became distinctly positive with LA45 within 24 h, with a maximum expression on days 3-6 in a range of 65 +/− 19% (Day3). The in vivo study of activated cells from two of three patients with infectious mononucleosis showed LA45 positivity in CD3, CD8, and HLA class-II negative cells. One patient showed 84% LA45 positive T-cells, and the other showed 14% LA45 positive T-cells. These observations support the contention that expression of Face-2 on the cell surface can be increased by cell activation.

Madrigal et al. [60] found expression of the LA45 epitope on EBV-transformed B cell lines as well as lectin-activated T cells, but not on long-term T cell lines or unstimulated peripheral blood T cells. The specificity of the LA45 antibody is precisely correlated with the sequence arginine-asparagine (RN) at residues 62 and 63 of the helix of the αl domain. The LA45 epitope is associated with many alleles of both HLA-A and -B loci but not those of the HLA-C locus. All the results are consistent with the presence of pools of free HLA-A and -B HCs at the surfaces of some but not all cell types.

### 2.3. Novel Cold Trypsin Assay Eliminated Face-2 from Cell Surface

Demaria et al. [61] studied the induction of Face-2 on activated human peripheral blood T cells, which were stimulated with phorbol myristate acetate (PMA) in different periods of time. PMA induced the expression of Face-2 on 90% of the cells stained with the mAb HC 10. The expression of Face-2 was detected on the surface of PMA-activated T cells by 8 hr, prior to expression of the interleukin-2 receptor (IL-2R), and reached plateau levels after 12 hr. Similarly, Face-2 is expressed on T cells activated with anti-CD3 or with PHA. On average, 30 to 65% of T cells were found to express Face-2 after 18 hr of stimulation. In addition, Face-2 is also expressed on the CD4+ T cell clone 4D9 and detected on the cell surface of activated T cells after brefeldin A (BFA) treatment, which arrests the exit of newly synthesized Face-1 from the ER. Expression of Face-1 was completely inhibited by BFA, and thus induction of Face-2 on activated T cells requires the transport of newly synthesized Face-2 to the surface. Interestingly, the expression of Face-1 molecules was up-regulated by PMA, as evidenced by an increase in the staining intensity of cells with mAb W6/32. This up-regulation was similarly abrogated in the presence of BFA.

To further investigate the relationship between the two Faces of HLA-I molecules, Face-2 heavy chains were eliminated from the cell surface with trypsin. The trypsin assay involves direct incubation with 25 pg/mL of trypsin in RPM1 1640 for 15 min at 4 °C [61]. Their subsequent re-expression was examined. Since high levels of Face-2 are expressed on all HTLV- l+ and EBV-transformed cell lines, HUT- 102 leukemia cells as well as PMA-activated T cells were briefly treated with cold trypsin. Trypsinization of these cells removed cell surface Face-2 but not Face-1 molecules. Interestingly, BFA failed to prevent the reappearance of Face-2 on activated T cells following their selective removal with trypsin, suggesting that (1) the generation of Face-2 is a continuous process and (2) Face-2 may be generated independent of Face-1. Aiuti et al. [62] using purified or biotinylated mAb L31, recognizing only Face-2 but not Face-1 of HLA-C, have shown (1) that the association with β2m is not necessary for the expression of HLA-C on the surface of activated T lymphocytes and (2) that cell activation affects the balance between the two conformational forms (Face-1 and -2) of HLA-C.

### 2.4. HLA-I Polyreactive mAbs Binding to Cryptic Sequences Exposed on Face-2

Face-2 molecules expressed on the cell surface are a “naked version” of HLA as they are exposing otherwise cryptic domains of amino acid sequences or epitopes. Therefore, Arosa et al. [63] designated Face-2, as Open Conformers [64]. Those amino acid sequences that are cryptic when HLA occur as quaternary structures, upon transformation to the monomeric version, become a target for immune recognition, due to exposure of the cryptic domains. Table 2 presents several epitopes exposed on Face-2 of HLA-I alleles. Some amino acid sequences are commonly shared with all other isoforms (e.g., ^117^AYDGKDY^123^, ^126^LNEDLRSWTA^135^ & ^137^DTAAQI^142^) (Table 2, Figure 4A). These epitopes may elicit IgG antibodies that are reactive with multiple HLA-I isomers, and hence designated as polyreactive HLA antibodies. These antibodies are each specific for a given epitope that may be found on numerous alleles of HLA-I isoforms. Hence, the antibody may be reactive with numerous alleles that express the particular (public) epitope, and in this fashion is polyreactive.

The degree of polyreactivity of an antibody depends on the presence of an amino acid sequence (epitope) on different alleles of different isoforms. Several factors may play a role in the recognition of a “polyspecific” epitope. This is well illustrated in Table 2 and Table 3. The sequence ^117^AYDGKDY^123^ is found in almost all HLA alleles of different isoforms. However, the same sequence with two additional amino acids ^115^QFAYDGKDY^123^ is not found in all HLA alleles of different isoforms as illustrated in Table 2. The difference between these two peptides is clarified by the antigenicity rank of epitopes, which is predicted by various methods to measure beta-turn, antigenicity, flexibility and hydrophilicity (Table 3)**.**
Figure 4A illustrates the exact location of the amino acids QF in the epitope. The methods predict the probability of specific sequences in an allele to antibodies. This finding stresses the importance of the role of each amino acids in promoting antigenicity, immunogenicity, and ultimately the polyreactivity of the antibodies they generate.

This type of polyreactivity is distinct from the polyreactivity of “natural” antibodies in which a single antibody may be reactive with multiple different epitopes. Antibodies against Face-1 epitopes may be shared among multiple alleles (e.g., Bw4), which are referred to as “public epitopes”. The polyreactive antibodies formed against the cryptic and commonly shared epitopes of Face-2 are exposed due to activation of cells during inflammation, infection, surgery and cyto/chemokine upregulations [58,59,60,61,62,63,64]. These polyreactive antibodies, are directed against Face-2 specific epitopes and confound the proper assessment of truly allograft-pathogenic DSA directed against epitopes on Face-1. In fact, several clinical investigators have recognized that not all HLA-I antibodies are detrimental to allografts [29,64,65,66,67,68,69]. Identifying “the real” pathological HLA antibodies is critical to protect the allograft after transplantation.

Pathological HLA antibodies are directed against the sequences exposed on the α1 and α2 domain at or near the peptide binding groove of Face-1 of HLA. In contrast to polyreactive antibodies, the monospecific antibodies bind only to an epitope unique to one or two alleles of an HLA isoform. These monospecific anti-Face-1 antibodies may be directed against a specific epitope, which is often referred to as a “private epitope” when restricted to a single allele or serological antigen. A typical example is mAb TFL-033 [55], which binds only to specific epitopes (e.g., ^65^RSARDTA^71^ and ^143^SEQKSNDASE^152^) found on specific alleles such as HLA-E^R101^ or HLA-E^G101^ (Table 2, Figure 4B)

### 2.5. Admixture of Face-2 with Face-1 on Beadsets Hampers Specific Monitoring Pathogenic Anti-Face-1 HLA Antibodies in Transplant Patients

We [52] have identified that two of the mAbs (TFL-006 and TFL-007) produced after immunizing the Face-2 of HLA-E are highly reactive to the most commonly shared peptide sequence (^117^AYDGKDY^123^) of HLA-I isoforms illustrated in Figure 4A. These amino acid sequences, when mixed with mAbs TFL-006 or TFL-007, inhibit the binding of the mAbs to Face-2 HLA-I coated on a solid matrix (SABs). Such polyreactive HLA-IgGs are capable of binding to Face-2 (open conformers) on the cell surface or on the beadsets. While investigating the HLA-E reactivity in alloimmunized normal males, polyreactive HLA-Ia reactivity was observed in anti-HLA-E-positive sera. These HLA-I polyreactive antibodies that are also formed during end stage organ diseases are particularly important from the perspective of HLA sensitization that occurs in patients with such diseases [70,71]. HLA allo-antibodies formed against allograft specific alleles that are present in the donor and not in the recipient can cause damage to the transplanted allograft. However, the polyreactive HLA-I IgGs capable of reacting only with Face-2 of several HLA isoforms and their alleles will not react with Face-1 expressed on allograft endothelial cells. Therefore, the polyreactive IgGs may not be the ones involved in antibody-mediated destruction of allografts [70,71].

Dr. Adam K. Idica, a dynamic researcher from Terasaki Foundation Laboratory has developed a category of beads that contain only Face-1 but not Face-2, based on the trypsin assay of Demaria et al. [61] for eliminating Face-2 but not Face-1 from the cell surface. Utilizing a modified version of the assay, One Lambda Inc. has generated beads carrying only Face-1, called iBeads, which are different from regular LABScreen SABs, in which Face-1 is admixed with Face-2 [56,71]. The LABScreen beads were commercialized for monitoring HLA antibodies in pre- and post-transplant patients’ sera. The regular LABScreen beads are reactive to mAbs W6/32, HC-10 and the HLA polyreactive mAb TFL-006 [70]. Unlike regular LABScreen beads, iBeads reacted only with W6/32 but not TFL-006 [55,56]. Another vendor (Immucor, Norcross, GA, USA) has also developed multiplex beads for monitoring HLA antibodies using a SAB assay in Luminex flowcytometry. These beads are called LIFECODES. When tested for the presence of Face-2 HLA-I using mAbs W6/32, HC-10 and TFL-006, it was noted that this beadset is totally devoid of Face-2 (open conformers) of HLA [72], similar to iBeads, which have been abandoned by the vendor of LABScreen beadsets.

One of the fundamental questions in measuring antibodies against HLA class-I or -II antigens is whether the assay detects the antibodies directed only against Face-1 (intact or native HLA; closed conformers) or against both Face-1 and Face-2 (monomeric version or open conformers). This is critical because numerous studies have documented that antibodies against intact HLA but not those formed against Face-2 are pathogenic (59). There is an imminent need to provide clinicians and HLA laboratories with beadsets coated only with Face 1 (closed conformers) and not admixed with Face-2 (open conformers) of HLA. Therefore, we have undertaken [72,73] detailed studies on the sera of end stage renal disease patients confirming that LABScreen-beadsets (LS) are coated with both Face-1 and Face-2 versions of HLA-I antigens, while LIFECODES-beadsets (LC) are devoid of Face-2. Serum IgG antibodies of the end-stage renal disease patients from two different cohorts [72,73] were examined for allo-HLA reactivity to the SABs from both vendors. The results confirm significant inter- and intraindividual variability in the number and strength of HLA antibodies monitored using SABs from the two vendors. Sera that reacted strongly with LABScreen SABs (>13,000 MFI) but weakly or not at all with LIFECODES SABs (<1000 MFI) gave negative T and B cell flow cross match (FCXMs). In contrast, sera that reacted with LIFECODES SABs (>13,000 MFI) but weakly with LABScreen SABs (<2100 MFI) exhibited positive FCXMs [56]. Thus, detection of antibodies directed against Face-2 in SAB assays may lead to inappropriate listing of unacceptable antigens, refusal of a potential donor, or unnecessary pre- or post-transplantation desensitization protocols.

### 2.6. Overexpression of Face-2 in Malignant Cells

Both Face-1 and Face-2 of HLA class-I molecules on the surface of cancer cells are specifically identified by the following antibodies: mAb W6/32 for Face-1 and for Face-2 by mAbs LA-45, LHC10, L31, and M38 [74,75,76]. Using mAb L31, Marozzi et al. [77] analyzed the Face-2 expression in neuroblastoma (NB) cell lines IMR-32 and LAN-1 and showed that cell surface expression of Face-2 is regulated post-translationally based on the following observations. These cell lines barely expressed Face-2 (mAb L31-positive) and Face-1 (mAb W6/32-positive) molecules in the basal state, but overexpressed Face-2 molecules upon differentiation with either retinoic acid or serum starvation. The expression was not accompanied by an increase of cell surface Face-1. In contrast, recombinant IFN-γ treatment led to the expression of Face- 1 on the cell surface on NB cell lines. No changes in the synthesis of either HLA-I and β2m mRNAs or of L31-positive Face-2 were observed in differentiated NB cells. Based on these findings, they conclude that the surface expression of Face-2 is regulated post-translationally.

Martayan et al. [78] examined both Face-1 and Face-2 expression in the renal carcinoma cell line KJ29, which is characterized by the loss of one copy of the β2m-gene, before and after transfection of β2m gene. They used three different mAbs to monitor the expression of the two Faces of HLA, they used L31(α1 of heavy chain of HLA-1), W6/32 (α2 region) and Q1/28 (α3 region). Transfection with β2m causes a coordinate change in the mAb reactivity of the three domains of HLA-C molecules. It was shown that the assembly with β2m affects the folding of not only the α1 and α2, but also of the α3 domain. The data demonstrated an impaired ability of Face-2 HLA-C to properly fold and largely remained as Face-2, even after transfection.

Giacomini et al. [79] have shown that the binding of mAb L31 depends on the presence of an aromatic residue at position 67. Essentially, the mAb L31 binds to all HLA-C alleles. The investigators tested L31 binding on several human tumor cells lines: HF-29 (colon carcinoma), MCF-7, SK BR-3, CPT-1, DAL-1 and VNT-1 (breast carcinoma), SK OV-3, (ovarian carcinoma), T24 (bladder carcinoma), AQB-1, Colo 38, SK Mel-37, SK Mel-39 (melanoma) and KJ29 (renal carcinoma). In addition, they have also tested snap frozen normal breast tissues and several cancer tissues. Most importantly, the L31 reactivity was observed on both the cytoplasm and on the cell surface of these tumor cells. The exposure of viable cells (HT-29, colon carcinoma) to IFN-γ upregulated the expression of both Face-1 (W6/32 positive) and Face-2 (L31 positive) molecules on the cell surface.

Furthermore, two popular commercial mAbs (MEM-E/02 and 3D12) marketed as HLA-E specific, which has been refuted by extensive research [50,51]. The binding of MEM-E/02 is selectively inhibited by two of the shared peptides (^137^DTAAQI^142^ and ^115^QFAYDGKDY^123^) but not inhibited by the peptide sequences specific for HLA-E When mAb 3D12 reactivity [80] was re-evaluated, using LABScreen beads coated with Face-1 admixed with Face-2 of HLA-A, HLA-B and HLA-C alleles, it simulated the reactivity of MEM-E/02 [51]. Therefore, it is inappropriate to designate those mAbs as HLA-E specific. However, both MEM-E/02 and 3D12 were used with the intention of localizing HLA-E on the cell surface of breast cancer [81,82], ovarian and cervical cancer [83,84], melanoma [85,86], glioblastoma [87,88,89,90,91], laryngeal carcinoma [92], colorectal carcinoma [93,94,95], renal cell carcinoma [96,97], and lymphoma [98]. Using HLA-E (Face-2) polyreactive antibodies may lead to errors in interpreting the results of immunodiagnosis of HLA-E on tissues and cancer cells. Since Face-2 is recognized as an important component of cancer cells, these polyreactive mAbs can never be diagnostic tool for tumor-associated HLA-E.

A mAb used for the immunodiagnosis of a specific allele should be monospecific and should be verifiable by blocking the binding with the allele specific epitopes or peptides (Figure 4B) see also [55]. We [55,99] have generated and identified an HLA-E monospecific monoclonal antibody (TFL-033). Identifying the epitopes unique to HLA-E, and dosimetric inhibition using the HLA- E specific epitopes, we could confirm the HLA-E monospecificity TFL-033 [55,99]. In contrast to MEM-E/02 and 3D12, TFL-033 is not reactive with any of the HLA class I alleles other than HLA-E. Comparing the HLA-E monospecific mAb TFL-003 and HLA-I isoforms-polyreactive MEM-E/02, Sasaki et al. [100] immunostained the cell surface of different types and stages of gastric cancer, as summarized in Figure 5. These observations illustrate the expression of both Face-1 (HLA-E) and Face-2 (HLA-I isoforms) in gastric cancer.

Evidently, activated tumor cells (with metastatic potential) may over express Face-2. Such exposure may result in antibody production against Face-2. We have documented that the Face-2 (or open conformer) polyreactive IgG2a mAbs TFL-006 and TFL-007 [52,53,54] are capable of suppressing IgG production by CD19+ B cells and proliferation of CD4 T cells [54,100,101]. The natural occurrence of such immunosuppressive anti-Face-2 antibodies may promote tumor progression and metastasis and anti-Face-2 mAbs can serve as valuable tool for passive immunotherapy of human cancers.

### 2.7. Homodimerization of HLA-I Face-2: The Face-3

That the Face-2 molecules expressed on the cell surface have the propensity to interact with other Face-2 molecules was first proposed by Arosa et al. [64]. The available literature documents that the cysteine residues exposed in Face-2 may contribute to this interaction resulting in Face-3 molecules (homodimers if both HCs are from the same allele). Clear evidence is available with reference to HLA-B27, HLA-C and HLA-G (*vide infra*).

### 2.8. Face-3 of HLA-B27

HLA-B27 is associated with the development of common inflammatory rheumatic diseases such as the spondyloarthritides (SPA), ankylosing spondylitis (AS), and reactive arthritis [102]. HLA-B27 is also associated with inflammatory eye disease, anterior uveitis [103]. HLA-B27 is devoid of β2m and HLA-II in a transgenic mouse model that develops arthritis. There are two major characteristics of HLA-B27 that distinguished it from other HLA Face-1 molecules. One, it has a tendency to misfold [104,105] and appear as Face-2 molecules. The other, the free HC (Face-2) in B27 has an unpaired cysteine residue (Cys^67^) in α1, just above the peptide binding domain. This unique feature of Face-2 of B27 leads to the finding that the cysteine carrying Face-2 molecules bind with peptide [46,106,107,108], and have a propensity for forming heavy chain homodimers (Face-3) on the cell surface [106,109,110]. The newly synthesized HLA-B27 HCs oligomerized in disulfide-linked complexes, including dimer-sized structures. The misfolded HLA-B27 oligomers accumulated in HeLa cells, even in the presence of an intact class-I assembly pathway. Dangoria et al. [104] showed that the formation of the homodimer was dependent on the unpaired cysteine at position 67 (Cys^67^).

Allen et al. [106] reported that the Face-2 of HLA-B27 can homodimerize to form Face-3 HLA-B27. While the Face-2 of HLA-B27 is recognized using mAbs HC-10 and ME1 but not by using W6/32, the homodimer (Face-3) was recognized on the cell surface by W6/32, particularly in the absence of β2m. Using surface labeling and immunoprecipitation, it was shown that in the absence of β2m, HLA-B27 heavy chain homodimers (termed HC-B27) were stabilized with a known peptide epitope. The “HC-B27” is a typical example of Face-3.

Naturally occurring cell surface Face-3 (HLA-B27 homodimers) are hypothesized to be recognized by killer immunoglobulin receptors (KIR, such as KIR3DL2) in the leukocyte immunoglobulin-like receptor family (LILR), and trigger inflammation [110]. The splenocytes (primarily T and B lymphocytes) from transgenic rats accumulate significant amounts of HLA-B27 heavy chain monomers (Face-2) and disulfide-linked oligomers (Face-3) [111]. Antoniou and others [112] documented that the ability to form homodimers (Face-3) is not restricted to HLA-B27. Homodimerization can be induced in other HLA-I molecules by slowing the rate of egress from the ER (e.g., by culture at 26 °C) and can be mediated by cysteines other than Cys67, including Cys164. The assembly kinetics of HLA-B27 in the ER is slower than many other HLA-I molecules, primarily as a result of the specific residues forming the peptide-binding groove. This increase in the time taken for the heavy chain to fold and exit the ER appears to contribute to the accumulation of misfolded ER-resident molecules, which leads to ER stress and the initiation of the unfolded protein response (UPR) [111].

HLA-B27 contains a relatively rare unpaired cysteine at position 67 in the peptide-binding groove, which has been implicated in the formation of these cell surface Face 3 HC dimers. These are generated via recycling of unstable fully folded HLA -I heterotrimers on the cell surface of a variety of cells such as human lymphoid cell lines [109], as well as on stimulated lymphoid cells from B27^+^ transgenic rats and on mononuclear cells of synovial and peripheral blood of B27^+^ patients with ankylosing spondylitis [110,113,114]. Additionally, it was documented that in both the dendritic-cell-like cell line KG1 transfected with HLA-B27 and in primary human monocyte-derived dendritic cells (DC) from HLA-B27^+^ individuals, dimers can be detected upon DC maturation [115]. HLA B27 homodimers on the surface of T-cells, monocytes, and natural killer cells may trigger innate immune mechanisms through binding to KIRs [116].

HLA-B27 expresses four unpaired cysteines (C) at positions 67, 101, 164, and 203 (Table 4). Allen et al. [106] and Bird et al. [109] showed that C67, located within the antigen binding groove, participated in the dimerization of HLA-B27 heavy chains, of which there are two populations: one is on cell surface and the other is ER-resident, and they are formed independently. In addition, HLA-I heavy chains possess four conserved C residues, which form two structurally important disulfide bonds within the α2-domain between C101–C164 and the α3-domain between C203–C259. Lenart et al. [117] showed that the structurally conserved cysteines 101 and 164 do participate in HLA-B27 dimerization. Using reducing and oxidation reactions, they demonstrated that HLA-B27 forms conformationally distinct dimeric species. Further analysis of HLA-B27 dimers indicated a nonhomogenous population involving C164 in ER HLA-B27 dimer formation. They further documented that the non-availability of both C164 and C67 during misfolding may also enable dimerization via C101.

### 2.9. Face-3 of HLA-G

HLA-G exhibits limited polymorphism. Its expression is restricted to the invasive extravillous trophoblasts of the decidua, and to tumor cells. Conventional HLA-G is expressed on the cell surface as β2m-associated HCs or Face-1. Comparison of the sequence of the HLA-G molecule to other HLA-I proteins revealed two unique cysteine residues located in positions 42 and 147. Boyson in the Strominger team [118] demonstrated that HLA-G forms disulfide-linked homodimers that are present on the cell surface. Immunoprecipitation of HLA-G revealed the presence of a 78-kDa form of HLA-G HC that was reduced by using DTT to a 39-kDa form. Mutation of Cys-42 to serine-42 completely abrogated dimerization of HLA-G, suggesting that the disulfide linkage formed exclusively through this residue. These findings were confirmed by other investigators as well [119].

### 2.10. Heterodimerization of Two Different HLA-I Heavy Chains: Face-4

It is also evident from the above reports that the open conformers or Face-2 have exposed cysteine residues in α1 and α2 helices that contribute to homodimerization of two Face-2 molecules. This raises several questions: Do the Face-2 cysteine residues of two open conformers get united in the endoplasmic reticulum, or post-translationally on the cell surface, or during cell activation by cytokines, or during capping or aggregation of Face-2 molecules on the cell surface? Lenart et al. [117] have shown that cysteine residues located at position 101 and 164 could also contribute to homodimerization. The cysteine residues are located at different positions in HLA-I isoforms as shown in Table 4. Cysteine at position 67 is found in several B alleles. Theoretically, all Face-2 HLA-I isoforms are capable of homodimerization since all of them possess cysteine. The question still remains whether dimerization occurs between Face-2 of the same allele or between two different alleles. Essentially, the presence of Face-2 implies the exposure of the cryptic sequences. The epitopes or eplets (3–5 amino acids) are not only recognized by antibodies and other cell surface ligands overexpressed upon cell activation, but also such interaction may occur between two different alleles expressed as Face-2 on the cell surface. We hypothesize that the linkage may not be restricted to cysteine residues but may also involve salt linkage, H-bonding, and van der Waal forces.

Although heterodimerization of two different Face-2 alleles has not been extensively studied, an analytical investigation on HLA-F renders promise for the existence of Face-4 of HLA. Although HLA-F is found intracellularly, it is not expressed in the majority of cells. Cell surface expression of HLA-F is detected on B cells and monocyte cell lines and in vivo on extravillous trophoblasts into maternal decidua [120,121,122,123]. Though intracellular HLA-F expression was observed in all resting lymphocytes, the surface expression was upregulated upon activation of B cells, T cells, NK cells and monocytes [120,121,122,123]. Goodridge and others [124] demonstrated that any of HLA-I HCs would interact with HLA-F only when the latter was in the form of an open conformer. This interaction was directly observed by coimmunoprecipitation and by surface plasmon resonance and indirectly on the surface of cells through coincident tetramers and HLA-I Face-2 colocalization. The data indicate that HLA-F is expressed as Face-2, and it has physical interaction with HLA-I heavy chains (Face-2). Furthermore, based on these data and the differential coimmunoprecipitation experiments, it appears likely that more than one complex form of HLA-F is expressed, with some or all complexed with HLA-I HCs. Taken together, this study provides evidence of at least three different forms of HLA-F based on differential staining of surface HLA-F using mAbs 4A11 and 3D11 over the course of lymphocyte activation and the unique 4B4 binding pattern. The authors also speculate that “HLA-F and MHC-I HC interactions can occur *in trans* between cells” (p. 6206).

Interaction of the HCs of various HLA-I isoforms with HLA-F HC is not surprising because of the possibility of HLA-I HCs to dimerize with the polypeptides of insulin and epidermal growth factor receptors on the cell surface [125,126,127]. Can ‘’non-MHC heterodimer’’ be considered as the Face-5 of HLA, since it may differ in biological function [125,126,127,128]? Reviewing the emerging roles of HLA-F in immune modulation and viral infection, Lin and Yan [129] categorized functional roles for different faces of HLA-F. Face-1 of HLA-F is shown to be recognized by the inhibitory receptors immunoglobulin (Ig)-like transcript receptor 2 (ILT2) and ILT4) (130). Face-2 of HLA-F can be recognized with the highest affinity by the activating receptor KIR3DS1(128). Face-3 and Face-4 may interact with NK cell and T-cell receptors. More investigations are needed to confirm these interactions. However, caution should be exercised while testing these interactions between Face-2 and other molecules. Generating acid or thermally denatured Face-2 on beadsets [130] for evaluating these interactions could be far from reliable. Assessment of these interactions with naturally expressed Face-2 molecules such as that of HLA-B, HLA-C and HLA-F would be highly valuable.

## 3. Significance of the Faces of HLA-I

Each of us have six pairs of HLA class-I isoforms, HLA-A, HLA-B, HLA-C, HLA-E, HLA-F and HLA-G on the surface of immune and other cells. The allelic forms of these isomers can be identified using the sequence-specific oligonucleotide (SSO) and sequence-specific primer (SSP) technologies [131,132]. HLA laboratories in the US are required by the United Network for Organ Sharing (UNOS) to report HLA-A, -B, -C, Bw4, Bw6, -DRB1, -DRB3/4/5, −DQA1, -DQB1, -DPB1. Therefore, the research laboratories focus almost entirely on Face-1 of HLA isoforms. However, the report of Schnabl et al. [58] indicating that the activated T-cells express open conformers (Face-2) is a turning point on understanding the other Faces of HLA. New findings on Face-2 began to emerge, particularly pertaining to cysteine residues at position 67. In addition to the relevance of C67 to genesis of arthritis and uveitis, it leads to our understanding of the HLA-I homodimers (Face-3). Again, since HLA-B27 carried C67, the observations on homodimerization was restricted to B27. Further investigations may reveal the genesis of Face-3 and Face-4, since several alleles of other HLA isoforms also express C in the open conformers (Table 4). Although pairings of these cysteines between two Face-2 molecules were observed in B27, the pathobiology of these pairings is still unknown. For Face-2 molecule interactions to occur, other forces such as H-bonding, Salt linkage, and van der Waal forces could be involved. All aspects of homodimerization of Face-2 leading to the genesis of new HLA molecules are still in the early stages of research. It is often envisaged that the dimerization occurs in the endoplasmic reticulum. However, there is a strong possibility that the homo- or even heterodimerization can occur on the cell surface of activated cells. One possible reason is the overexpression of Face-2 of six pairs of HLA isoforms on the cell surface. Evidence from the studies made on HLA-F suggests that Face-2 can pair with its own allele or other HLA-I Face-2 molecules. Paucity of such observations and information on other HLA isoforms emphasizes the need for more research.

This is particularly important since Face-2, Face-3, and Face-4 could be considered as novel antigens. These Faces of HLA-I are of great interest in generating antibodies against diversified epitopes, generated by either monomerizaton or homo- and heterodimerization of the HLA heavy chains. The polyreactive antibodies (TFL- 006, and TFL-007) generated by Face-2 are capable of suppressing blastogenesis of CD4+ T cells and proliferation and antibody production of B cells [54,100,101]. In addition, Face-2 of HLA-E generated monospecific mAbs (TFL-033) that can bind to Face-1 of HLA-E specifically. The NK cell cytotoxic capability is lost when tumor-associated HLA-E binds to the CD94/NKG2A receptor, resulting in tumor progression and reduced survival. TFL-033 binding to HLA-E may prevent interaction with receptors (CD94/NKG2a) of NK cells and promote tumor cytotoxicity. Indeed, TFL-033 is capable of upregulating CD8+ T lymphocytes for antigen-specific tumor killing [55,133].

Carrido [134] has examined extensively the loss of HLA class-I on human cancer cells during tumorigenesis and metastasis. This review favors the assumption that the paucity of Face-1 on human tumor could also be due to transformation of Face-1 to Face-2, Face-3, and Face-4. These transformed faces of Face-1 HLA-I molecules are indeed novel antigens and serve as the primary targets for passive and active specific immunotherapy of many cancers [135] One such well known novel antigen is O-acetylated sialic acid on gangliosides, which are used as target antigens for an allogenic melanoma vaccine [136,137]. Face-1, Face-2 and Face-3 can be incorporated onto liposomes [138,139] or extra cellular vesicles to be used as vaccines. Klareskog et al. [140] have developed a protocol to making HLA-containing liposomes, which would be valuable to develop HLA-liposome vaccines to target the novel faces of HLA. More investigations are needed to fully understand the functional roles of these new faces of HLA-I in human diseases.

## 4. Conclusions

In this review, HLA-I molecules are classified into four categories (Face-1, Face-2, Face-3, and Face-4). Natural cell-surface Face-1 is glycosylated, in contrast to the recombinant Face-1 coated on solid matrices used to monitor anti-HLA antibodies. Cells activated during inflammation by chemo- and cytokines, the cells may express β2m-free monomeric HCs (Face-2). Anti-H polyclonal serum and mAbs LA45, L31, TFL-006, and TFL-007 recognize Face-2. Importantly, Face-2 is commonly overexpressed in malignant cells, and therefore, they may serve as an important target for passive and active specific immunotherapy or vaccines. Face-2 exposes cryptic sequences masked by β2m in Face-1. Often these sequences are shared among different isoforms and their alleles, referred to as public epitopes. The public epitopes serve as targets for immunoregulatory therapies such as IVIg therapy. They elicit antibody responses which are polyreactive. Such polyreactive antibodies occur in non-alloimmunized normal males. Recent investigations document the presence of pathogenic anti-Face-1 antibodies as well as non-pathogenic anti-Face-2 antibodies in end stage organ diseases and in cancer and transplant patients. The emergence of Face-1 specific iBeads or Immucor (LIFECODES) beads serves to distinguish anti-Face-1 antibodies from anti-Face-2 antibodies. Suppression of blastogenesis and proliferation of CD4+ T cells by highly polyreactive anti-Face-2 mAbs (e.g., TFL-006 and TFL-007) point out that they can be potential immunotherapeutic agents for controlling tumor progression.

The homodimerization of Face-2 generates Face-3 molecules. This interaction involves cysteine residues in HCs. Face-3 of HLA-B27 is associated with inflammatory rheumatic diseases. Face-3 of HLA-G expression is restricted to the invasive extravillous trophoblasts of the decidua, and to tumor cells.

Face-4 is a consequence of dimerization of Face-2 of two different alleles or isomers and is frequently observed on the cell surface. HLA-F and HLA-I HC interactions can occur *in trans* between cells. Naturally occurring cell surface Face-3 is recognized by KIRs (such as KIR3DL2) and may trigger inflammation. The different Faces of HLA-F are also recognized by the inhibitory receptors 2 (ILT2) and 4 (ILT4). Face-2 of HLA-F can be recognized with the highest affinity by the activating receptor KIR3DS1. Face-3 and Face-4 may interact with NK cell and T-cell receptors. While Face-1 is primarily involved in antigen presentation, the other faces of HLA-I may interact with inhibiting and activation receptors associated with inflammation.

Evidently, the Face-3 and Face-4 formed because of formation of homodimers and heterodimers is bound to generate novel conformationally altered epitopes. Identifying these epitopes will open a new chapter in vaccine-mediated immunotherapy of cancer and other diseases. Thus these faces of HLA may vary with various pathological conditions such as inflammation, malignancy and transplantation. Inter- and intra-cellular molecules of these faces could also improve our understanding of disease physiopathology. These faces of HLA could be proposed as excellent and practical targets for immunodiagnosis of cancer and other diseases and could serve as valuable novel targets for vaccine-mediated active specific immunotherapy.

Different faces of HLA-I have not been fully explored. Lack of monospecific antibodies is a significant issue. Today more than ever, immunologists need to identify the novel faces of HLA and their monospecific mAbs to refine both diagnosis and therapy.

## Figures and Tables

**Figure 1 vaccines-10-00339-f001:**
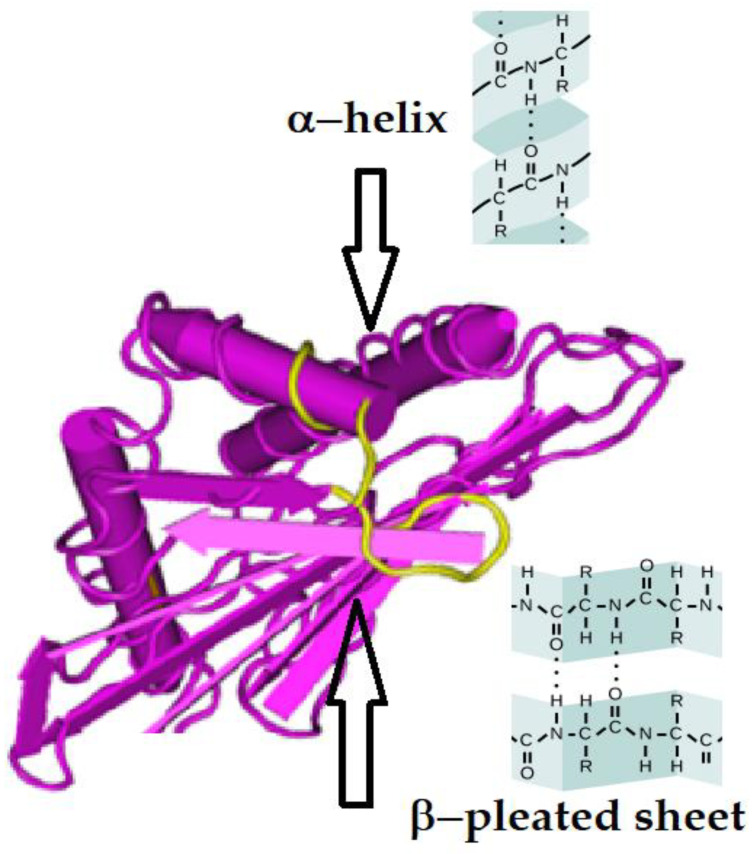
Three-dimensional structure of the HLA class-I heavy chain without β2-microglobulin showing the molecular arrangement of α-Helix and β-pleated sheets. The yellow strand of the protein chain represents the region of the cryptic epitope exposed in the absence of β2-microglobulin. The α3-helical domain is not shown in the figure.

**Figure 2 vaccines-10-00339-f002:**
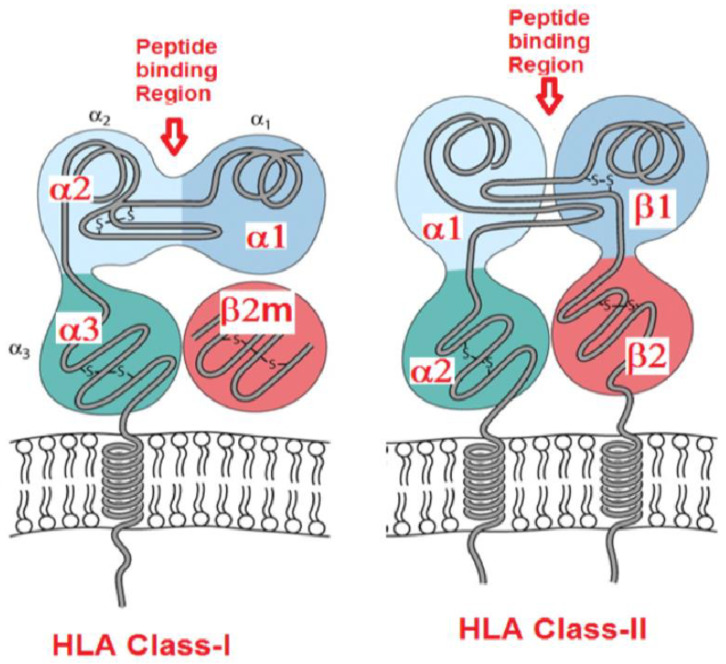
Diagrammatic illustration of the quaternary heterodimeric structure of HLA class-I with monomeric heavy chain with β2-microglobulin compared with the heterodimeric structure of HLA class-II with two heavy chains. This figure of HLA-II illustrates how the two heavy chains arrange themselves during dimerization. HLA-I heavy chain consists of α1, α2 and α3 helices. The peptide binding groove is located in between α1 and α2 helices of the single heavy chain of HLA-I, whereas the peptide binding groove is a consequence of a cleft formed by α1 and β1 helices of the two heavy chains of HLA-II.

**Figure 3 vaccines-10-00339-f003:**
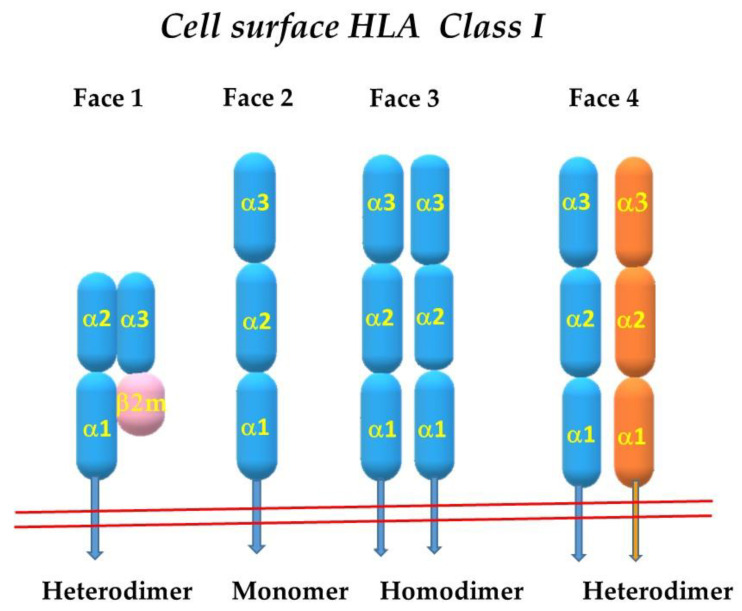
Diagrammatic illustration of the four different Faces of HLA class-I. **Face-1** is the conventional cell surface HLA-I formed by dimerization of heavy chain with β2-microglobulin, hence referred to as heterodimer or closed conformer. The peptide binding grove is located in between α1 and α2 helices of the single heavy chain of HLA-I. **Face-2** consists of heavy chain without β2-microglobulin. It is also known as open conformer. The figure illustrates linearized version but it is oriented conformationally. **Face-3** is a consequence of homodimerization of two heavy chains of the same alleles, while **Face-4** is a result of dimerization of two heavy chains of different alleles of HLA-I isomers.

**Figure 4 vaccines-10-00339-f004:**
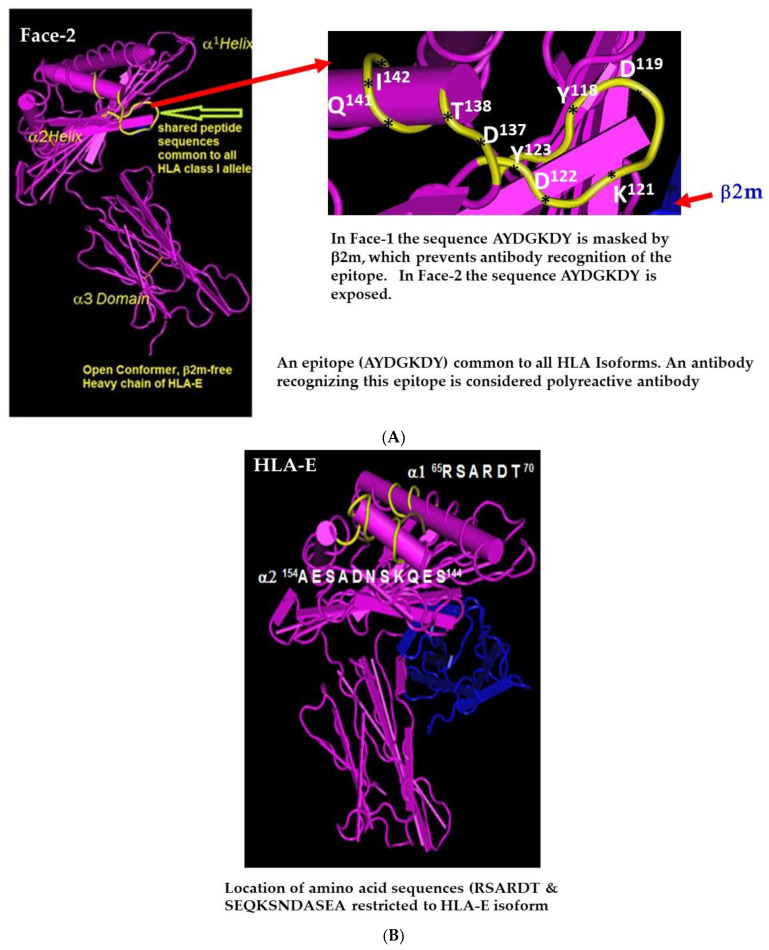
(**A**) Exposed cryptic peptide sequences of an HLA-I devoid of β2 in Face-2. Sequence (^117^AYDGKDY^123^) is the most commonly shared with other HLA alleles of five major isoforms is indicated in yellow (see also Table 2). An antibody binding to such an epitope shared by other HLA isoforms are called polyreactive antibodies. (**B**) Face-1 of HLA-E. HLA-I monospecific antibodies (mAbs) recognize epitopes exposed on α1 or α2 helix of an allele of an isoform. Monospecificity of a mAb (TFL-033) is illustrated with HLA-E restricted amino acid sequence RSARDT. This sequence is not shared with any other HLA isoforms. Dosimetric inhibition of the antibody binding to HLA-E with the specific sequence confirms the monospecificity of the mAb. Only monospecific mAbs are reliable for immunodiagnosis of HLA antigens expressed on human normal and cancer cells. A donor specific antibody is expected to be monospecific for the donor allele and not polyreactive with a sequence exposed on Face-2.

**Figure 5 vaccines-10-00339-f005:**
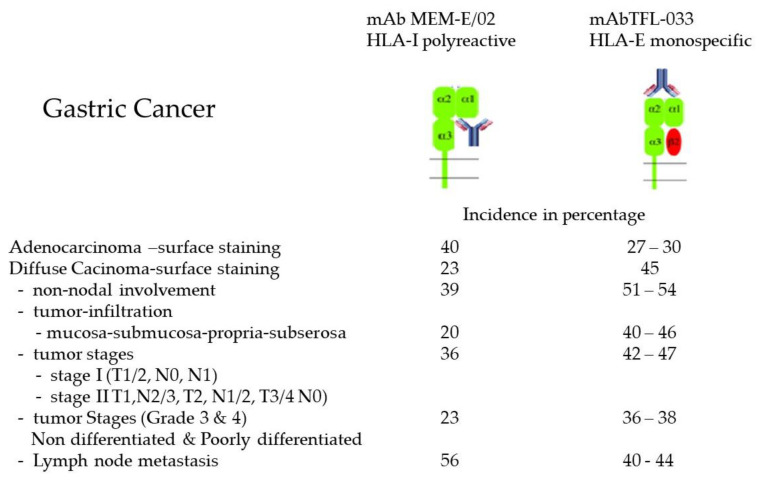
Face-1 (HLA-E) and Face-2 (HLA-I) expression on the cell surface of adenocarcinoma and diffuse carcinoma. Note high expression of Face-2 identified with MEM-E/02 in the lymph node metastatic cancer cells.

**Table 1 vaccines-10-00339-t001:** The number of named alleles for each gene of HLA-I and HLA-II, located on the short-arm of Chromosome 6.

Updated on	HLA Class I Alleles			HLA Class II Alleles					
	α-Chain			α-Chain			β-Chain		
	Genes	Alleles	Proteins	Genes	Alleles	Proteins	Genes	Alleles	Proteins
14 October 2020	A	6766	4064	DRA	29	2	DRB1	2949	2015
7 July 2021		6921	4156		29	2	3801	3801	2620
14 October 2020	B	7967	4962				DRB2	1	0
7 July 2021		8181	5090					1	0
14 October 2020	C	6621	3831				DRB3	390	293
7 July 2021		6779	3927					404	301
14 October 2020	E	271	110				DRB4	194	129
7 July 2021		271	110					203	131
14 October 2020	F	45	6				DRB5	155	120
7 July 2021		45	6					163	125
14 October 2020	G	82	22				DRB6	0	0
7 July 2021		88	26					3	0
14 October 2020	α-chain pairs with β2-microglobulin						DRB7	2	0
7 July 2021								2	0
14 October 2020							DRB8	1	0
7 July 2021								1	0
14 October 2020							DRB9	6	0
7 July 2021								6	0
14 October 2020				DQA1	306	143	DQA2	40	11
7 July 2021					343	172		40	11
14 October 2020							DQB1	1997	1303
7 July 2021								2033	1324
14 October 2020				DPA1	258	107	DPB1	1749	1106
7 July 2021					298	135		1862	1180
14 October 2020				DPA2	5	0	DPB2	6	0
7 July 2021					5	2		6	3
14 October 2020				DMA	7	4	DMB	13	7
7 July 2021					7	4		13	7
14 October 2020				DOA	12	3	DOB	13	5
7 July 2021					12	3		13	5
				α-chain pairs with β-chain					

Source: IPD-IMGT/HLA DATABASE WEBSITE https://www.ebi.ac.uk/ipd/imgt/hla/about/statistics/ (accessed on 7 July 2021).

**Table 2 vaccines-10-00339-t002:** Peptide sequences of Face-2 of HLA-E shared with other HLA alleles of five major isoforms and some sequences are restricted to HLA-E and A or B or C restricted.

Face-2 Peptide Sequences or Open Conformers of HLA-E	HLA Alleles					
	HLA-Ia			HLA-Ib		Specificity
[Number of Amino Acids]	A	B	Cw	F	G	
^47^PRAPWMEQE^55^ [9]	1	0	0	0	0	A*3306
^59^EYWDRETR^65^ [8]	5	0	0	0	0	A-restricted
^65^RSARDTA^71^ [6]	0	0	0	0	0	E-restricted
^90^AGSHTLQW^97^ [8]	1	10	48	0	0	Polyspecific
^108^RFLRGYE^123^ [7]	24	0	0	0	0	A-restricted
^115^QFAYDGKDY^123^ [9]	1	104	75	0	0	Polyspecific
^117^AYDGKDY^123^ [7]	491	831	271	21	30	Polyspecific
^126^LNEDLRSWTA^135^ [10]	239	219	261	21	30	Polyspecific
^137^DTAAQI^142^ [6]	0	824	248	0	30	Polyspecific
^137^DTAAQIS^143^ [7]	0	52	4	0	30	Polyspecific
^143^SEQKSNDASE^152^ [10]	0	0	0	0	0	E-restricted
^157^RAYLED^162^ [6]	0	1	0	0	0	B*8201-restricted
^163^TCVEWL^168^ [6]	282	206	200	0	30	Polyspecific
^182^EPPKTHVT^190^ [8]	0	0	19	0	0	C-restricted

**Table 3 vaccines-10-00339-t003:** The difference between peptides ^115^QFAYDGKDY^123^ and ^117^AYDGKDY^123^ clarified by the bioinformatics analysis done with the Immune Epitope Database (IEDB) to predict antigenicity rank of epitopes. Chou and Fasman beta turn, Kolaskar and Tongaonkar antigenicity, Karplus and Schulz flexibility and Parker hydrophilicity prediction methods in IEDB were employed to predict the immunogenic rank of the two sequences. Note the striking difference in antigenicity, flexibility and hydrophilicity between ^115^QFAYDGKDY^123^ and ^117^AYDGKDY^123^ due to presence are absence of the two amino acids Q and F. This finding stresses the importance of the role of each amino acids in promoting antigenicity as well as immunogenicity.

Peptide Sequence Exposed in the Absence of β2-Microglobuin	Number of Positive HLA-I Alleles						Prediction SCORES				Immunogenicity Rank
	Classical HLA-Ia			Non-Classical HLA-Ib		Specificity	Beta-Turn	Antigencity	Flexibility	Hydro Philicity	
[Total Number of Amino Acids]	A	B	Cw	F	G		Chou & Fasman (1978)	Kolaskar & Tangaonkar (1990)	Karplus & Schulz (1985)	Parker (1986)	
^115^QFAYDGKDY^123^ [9]	1	104	75	0	0	Polyspecific	1.059	1.001	0.993	2.629/3.201	5
^117^AYDGKDY^123^ [7]	491	831	271	21	30	Polyspecific	1.204	0.989	1.061	4.243	1

**Table 4 vaccines-10-00339-t004:** Cysteine residues exposed in Face-2 of HLA isoforms, capable of interacting within Face-2 same alleles or Face-3 of other alleles of the HLA-I isoforms present in the same cell.

	Cysteine Position					
Alleles of HLA-I Isoforms	42	67	101	164	203	259
B *1401/2/3/4/5/6		C	C	C	C	
B *1509/10/66		C	C	C	C	
B *2701/2–17/19–36		C	C	C	C	
B *3801/2/4/5/8–15		C	C	C	C	
B 41*3901/3–7/9/12/14/15/18/19/22/24/26–37/41		C	C	C	C	
B *7301		C	C	C	C	
B *7803		C	C	C	C	
almost All C* alleles			C	C	C	
almost All A* alleles			C	C	C	
HLA-E			C	C	C	C
HLA-F			C	C	C	C
HLA-G	C		C	C	C	C

## Data Availability

Data are archived in Terasaki Foundation Laboratory in the laboratory division of Mepur H. Ravindranath.

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
