# Peer review of "Four Faces of Cell-Surface HLA Class-I: Their Antigenic and Immunogenic Divergence Generating Novel Targets for Vaccines"

_vaccines, 2022, doi:10.3390/vaccines10020339_

Round 1
Reviewer 1 Report
Manuscript by Ravindranath et. al. describes upto date knowledge in the field of HLA molecules. Topic is of immense interest, however, manuscript suffers from many deficiencies, making it unsuitable for publication in its current format. Manuscript needs much work, below I am mentioning a few points that should help authors to improve its quality.
- Please fix font size on affiliations
- Abstract needs much simplification and flow of thought. In its current form, it’s difficult to follow topics being discussed due to the lack of coherence. It has to be more succinct, concise, and clear.
- Also, figure 1 needs to be moved to the main text, abstracts usually do not need supplementary figures.
- Figures image quality are very poor and needs to be improved and properly referenced, if not generated by the authors.
- There is not enough explanation in the figure legends on what is being shown in the figures.
- There are huge issues in the overall presentation, authors are jumping from one topic to another topic without properly introducing and discussing them one by one, e.g. isoforms are randomly introduced at page 3, which is followed by a discussion of dimerizations.
- Page 3 “with other monomers of related category of proteins to become heterodimers “ authors need to explain what is related category?
- Authors need to provide concise and clear meaning to each topic, current writing style and explanations are too vague, defeating the whole purpose of a review (examples in points 6 and 7 above).
I hope this helps.
Author Response
We thank the reviewer for his evaluation of the manuscript and for considering that the manuscript “describes upto date knowledge in the field of HLA molecules” and the topic is of immense interest”.
We also thank the reviewers for suggesting rectification for correcting the deficiencies in the manuscript and enabling to make it suitable for publication. The reviewers points are highly invaluable for improving the quality of the manuscript. We have revised the following concerns of the reviewer
- Please fix font size on affiliations.
Corrected
- The abstract needs much simplification and flow of thought. In its current form, it’s difficult to follow topics being discussed due to the lack of coherence. It has to be more succinct, concise, and clear.
The abstract is now revised and simplified, concise, succinct, and clear. The thought process is arranged in a logical sequence.
- Also, figure 1 needs to be moved to the main text, abstracts usually do not need supplementary figures.
Figure 1 is not in the Abstract but is in the Introduction of the main text. It is needed in the preamble section of the introduction to make it more understandable.
- Figures image quality are very poor and needs to be improved and properly referenced, if not generated by the authors
We have redone the figures for better image quality and we have also deleted a figure, that is not highly relevant and could not be improved further.
- There is not enough explanation in the figure legends on what is being shown in the figures.
We have taken extensive efforts to revise the figure legends, as suggested by the reviewer.
- There are huge issues in the overall presentation, authors are jumping from one topic to another topic without properly introducing and discussing them one by one, e.g. isoforms are randomly introduced on page 3, which is followed by a discussion of dimerizations.
We understand the concern of the reviewer on this aspect. We have taken efforts to properly introduce and discuss the topic one by one in a logical sequence. Wherever necessary we have changed the titles to make it crisp, clear, and in a logical sequence.
- Page 3 “with other monomers of the related category of proteins to become heterodimers “ authors need to explain what is a related category?
We have revised the issue on Page 3 in page 6 (in red)
- Authors need to provide concise and clear meaning to each topic, current writing style and explanations are too vague, defeating the whole purpose of a review (examples in points 6 and 7 above).
We thank the reviewer for this suggestion, we have revised each topic and the writing style and provided clear explanations.
Reviewer 2 Report
The manuscript represents an interesting approach of MHC class I binding and antigen-binding and expression. There are some issues to consider in the manuscript
1.-The authors show the expression of Fase MHC class I in figure 2 and then discuss the importance of Face 2 and the different interactions. It is important to define the % expression of each species and the probability of soluble MHC class I. Table I can be confusing to the reader and should be withdrawn just by making a citation in the text.
2.-Figure 4 should be figure 3. A discussion of HLA F and HL G small molecules should be close to the figure.
3.-The date at the end of Table 2 should be updated and the link of the web page and visit should be stated.
4.- The authors should explain better in table 2, why the peptide length along with the sequences of class E are important . The difference between peptides 115QFAYDGKDY123 and 117AYDGKDY123 should be discussed in detail since it is crucial for binding specificity and cell response.
5.- Table 4 should be discussed further. There is no information on the experimental assay. Use supplemental files to show raw data.
6.- The importance of overexpression of Face 2 in tumours requires attention. Figure 6 is confusing and there are two exact ranges to compare, use colors to highlight the expression and describe the experiment better. Use supplemental files to show raw data.
7.- In table 4, position 67 should be discussed further in terms of the interaction.
8.- It is important to mention the limitations of the study which is partially discussed at the end. The involvement of KIR receptors can be crucial in the response to the tumour.
Author Response
The manuscript represents an interesting approach of MHC class I binding and antigen-binding and expression. There are some issues to consider in the manuscript
We thank the reviewer for considering that the manuscript represents an interesting approach of HLA-I expression and for favorable and reasonable evaluation as evidenced by the stars.
1.-The authors show the expression of Face MHC class I in figure 2 and then discuss the importance of Face 2 and the different interactions. It is important to define the % expression of each species and the probability of soluble MHC class I. Table I can be confusing to the reader and should be withdrawn just by making a citation in the text.
To define the percentage expression of each face is in HLA class I as a whole is an impossible task, for the following reasons:
- The expression of Face-2, Face-3, and Face-4 differ with each isoform such as HLA-A, HLA-B, HLA-C, HLA-E, HLA-F, and HLA-G.
- HLA-A is most restricted to Face-I and upon activation of cells develop Face-2. There is a published record of Face-3 and Face-4 for HLA-A.
- HLA-B is strikingly different in that it has cysteine at position 67 which is involved in homodimerization resulting in the formation of Face-2. B27 is a typical example and Face-3 in B27 is correlated with anchylosing Spondylosis and other autoimmune diseases. Cysteine residues occur in several positions in HLA-B suggesting the propensity to form S-S bonds during homo- or heterodimerization of Face-2 molecules.
- Previous publications point to overexpression of Face-2 in HLA-C. Although there is a propensity to form Face-3 and Face-4, more observations or publications are needed to evaluate the formation of other faces in HLA-C.
- HLA-F is quite interesting for Face-2 of HLA-F is more prevalent than Face-1. Evidently there the face-2 of HLA-F was to be shown to form Face-3 and Face-4.
- HLA-E and HLA-G do have Face-2 as cited in the literature.
- Most importantly, there are several publications to document the expression of Face-2 upon activation of immune cells due to viral infection, during carcinogenesis, upregulation of chemokines, and cytokines.
The very purpose of this review is to draw the attention of immunologists, particularly those involved in clinical transplantation, tumorigenesis, and inflammatory diseases and to create an awareness of multiple faces of HLA-I. Face-1 is not the only face of HLA. Anyone who has studied antibodies to HLA-I in normal healthy individuals or in patients with end stage organ disease, will realize that there IgG antibodies against not only one’s own HLA but also for tens of allo-HLA-I molecules. Often some such antibodies formed in transplant patients are mistaken to be Donor specific antibodies. Worst of all severe desensitization therapies are given by clinicians for the patients with end-stage organ disease with the belief that B cell suppression will lower the HLA-antibodies. Above al IVIg is given to the patients to lower HLA antibodies, without realizing that IVIg purified from plasma of tens of thousands of normal healthy donors, contain a high level of antibodies against HLA-I and even HLA-II. Therefore, there is an imminent need to create an awareness of the Faces of HLA.
2.-Figure 4 should be figure 3. A discussion of HLA F and HL G small molecules should be close to the figure.
We very much regret to note errors in the figure numbers. In fact, there are TWO figure 2s. We will correct them during revision. We will also bring the discussion of HLA-F and HLA-G closed to the figure. We have deleted a figure in the revised manuscript for it is not critical and its resolution could not be improved to show in a holistic perspective.
3.-The date at the end of Table 2 should be updated and the link of the web page and visit should be stated.
It is done in the revised manuscript.
- The authors should explain better in table 2, why the peptide length along with the sequences of class E is important. The difference between peptides 115QFAYDGKDY123 and 117AYDGKDY123 should be discussed in detail since it is crucial for binding specificity and cell response.
We thank the reviewer for this important comment and for emphasizing the need for clarification. In response to this comment, we have clarified the concerns of the reviewer with the following table, newly added as
Table 2B entitled The difference between peptides 115QFAYDGKDY123 and 117AYDGKDY123 clarified by the bioinformatics analysis done with the Immune Epitope Database (IEDB) to predict antigenicity rank of epitopes. Chou and Fasman beta-turn, Kolaskar and Tongaonkar antigenicity, Karplus and Schulz flexibility, and Parker hydrophilicity prediction methods in IEDB were employed. The methods predict the probability of specific sequences in HLA-E that bind to antibodies being in a beta-turn region, being antigenic, being flexible, and being in a hydrophilic region. Antigenicity rank is calculated by pooling the probability values. Note the striking difference in antigenicity, flexibility, and hydrophilicity between 115QFAYDGKDY123 and 117AYDGKDY123 due to the presence of are the absence of the two amino acids Q and F. This finding stresses the importance of the role of each amino acid in promoting antigenicity as well as immunogenicity.
In addition, we have added a separate paragraph on page 9 as follows:
The degree of polyreactivity of an antibody depends on the presence of an amino acid sequence (epitope) on different alleles of different isoforms. Several factors may play a role in the recognition of a “polyspecific” epitope. This is well illustrated in Tables 2A and 2B. The sequence 117AYDGKDY123 is found in almost all HLA alleles of different isoforms. However, the same sequence with two additional amino acids 115QFAYDGKDY123 is not found in all HLA alleles of different isoforms as illustrated in Table 2A. The difference between these two peptides is clarified by the antigenicity rank of epitopes, which is predicted by various methods to measure beta-turn, antigenicity, flexibility, and hydrophilicity (Table 2B). The methods predict the probability of specific sequences in an allele to antibodies. This finding stresses the importance of the role of each amino acid in promoting antigenicity, immunogenicity, and ultimately the polyreactivity of the antibodies they generate.
5.- Table 4 should be discussed further. There is no information on the experimental assay. Use supplemental files to show raw data.
Since the academic editor reports that the Table 4 contains data that is extremely specific and of little interest to a "Vaccines" review reader, we regretfully delete the table and restricted it to description and comments in the text.
6.- The importance of overexpression of Face 2 in tumors requires attention. Figure 6 is confusing and there are two exact ranges to compare, use colors to highlight the expression and describe the experiment better. Use supplemental files to show raw data.
We have revised this section on the overexpression of Face-2 in tumors. Following the recommendations of the reviewer, we have revised Figure 6 (Now Figure 5) using colors to highlight the expression and described the experiment and furthermore, we have clarified associated figures. Based on the Figure, we have stated “Figure 5 clearly documents that MEM-E/02, which claimed to be specific for HLA-E is no longer truly specific for HLA-E and it reactive with different HLA-I alleles.”
7.- In table 4, position 67 should be discussed further in terms of the interaction.
In the revised version, we have further discussed the dimerization involving position 67. Table 4 has become Table 3. In the paragraph above the Table, we have discussed further in terms of the interaction of C67. We have also discussed that the dimerization of B27 is not restricted to C67 but also may involve other conserved C101 and C164.
8.- It is important to mention the limitations of the study which is partially discussed at the end. The involvement of KIR receptors can be crucial in the response to the tumor.
We have taken note of the comment on KIR receptors and clarified it further in the revised version.
Reviewer 3 Report
Comment 1.
The authors deal with the issue about the neoantigens relating with only the four faces of cell-surface HLA Class-I in the manuscript. The authors should add another subsection in the manuscript that deals with how the cytoplasmic proteins with mutated amino acid residue are degraded into peptides, and how those peptides are transported to the cell surface, and bind cell-surface HLA Class-I, becoming the neoantigens.
Author Response
We thank the reviewer for considering that this work is “a significant contribution to the field”, “well organized and comprehensive”, “scientifically sound” and references are appropriate and adequate”.
As recognized by the reviewer, the work describes four different kinds of a molecule of cell surface HLA-I, which we designated as Faces 1 to 4. Not all four different kinds of HLA-I molecules are capable of presenting peptides. Only the trimeric HLA-I (designated as Face-I) present peptides, whereas the other three different HLA molecules are unique and as of today there is no clear evidence documenting their involvement in the presentation of any neoantigens. The functional aspects of the other three HLA-I molecules are involved in receptor-ligand interaction with various other cytoplasmic components, as described.
Round 2
Reviewer 1 Report
The manuscript has substantially improved, I am happy to have this accepted.
Author Response
We profusely thank the reviewer for accepting the revised version of the manuscript.
Reviewer 2 Report
The manuscript was corrected according to the suggestions. The responses to the queries are accepted. Overall an interesting manuscript
Author Response
We thank you for accepting the revised version of the manuscript and for considering our review as an interesting contribution.
We now replied to two major comments of the Academic reviewer. Hope it is better and improved.
Thank you once again,